# Long-term changes in body composition and their relationships with cardiometabolic risk factors: A population-based cohort study

**Zhaoyang Fan**[1], **Yunping Shi**[2], **Guimin Huang**[3], **Dongqing Hou**[3], **Junting Liu**[3]*

**1** Department of Early Childhood Development, Capital Institute of Pediatrics, Beijing, China, **2** Beijing Center for Disease Prevention and Control, Beijing, China, **3** Child Health Big Data Research Center, Capital Institute of Pediatrics, Beijing, China

* Junting_liu@163.com

**Data Availability Statement:** The datasets analyzed during the current study are available in

## Abstract

The aim of the present study was to classify the latent body fat trajectories of Chinese adults and their relationships with cardiometabolic risk factors. Data were obtained from the China Health Nutrition Survey for 3,013 participants, who underwent six follow-up visits between 1993 and 2009. Skinfold thickness and other anthropometric indicators were used to estimate body composition. The latent growth model was used to create fat mass to fat-free mass ratio (F2FFMR) trajectory groups. Blood pressure, fasting plasma glucose, total cholesterol, triglycerides, and high- and low-density lipoprotein–cholesterol were measured in venous blood after an overnight fast. Logistic regression was used to explore the relationships of F2FFMR trajectory with cardiometabolic risk factors. In men, four types of F2FFMR trajectory were identified. After adjustment for behavioral and lifestyle factors, age, and weight status, and compared with the Low stability group, the High stability group showed a significant association with diabetes. In women, three types of F2FFMR trajectory were identified. Compared to the Low stability group, the High stability group showed significant associations with diabetes and hypertension after adjustment for the same covariates as in men. Thus, in this long-term study we have identified three F2FFMR trajectory groups in women and four in men. In both sexes, the highly stable F2FFMR is associated with the highest risk of developing diabetes, independent of age and body mass. In addition, in women, it is associated with the highest risk of hypertension, independent of age and body mass.

## 1. Introduction

Obesity has become a major health problem [1], with the prevalence of overweight and obesity in adults reaching 36.9% and 38.0% in men and women, respectively [2]. Millions of deaths can be ascribed to obesity worldwide [3, 4]. Fat mass is a direct indicator for evaluating obesity, and obesity has trajectory effects [5], and longitudinal mixed-effects and latent growth curve models are commonly used to characterize changes in body mass index (BMI) and their

the following website: http://www.cpc.unc.edu/projects/china/.

**Funding:** Jt Liu received the fund of Beijing Hospitals Authority Youth Program, code: QML20191302(http://www.bjygzx.org.cn/). Zy Fan received the fund of National Key Research and Development Program of China, grant No. 2018YFC1002503(https://service.most.gov.cn/index/. The funders had no role in study design, data collection and analysis, decision to publish, or preparation of the manuscript.

**Competing interests:** The authors have declared that no competing interests exist.

relationships with subsequent outcomes [6–10]. Excessive fat mass has an adverse effect on cardiometabolic risk factors. However, there is a maximum capacity for adipose expansion, and when this is reached, lipids accumulate in other tissues and cause metabolic disease [11]. In contrast, fat-free mass protects against the development of cardiometabolic risk factors [12]. Therefore, we aimed to construct a metabolic load-capacity model, in which fat mass is the metabolic load and fat-free mass is the metabolic capacity [13, 14].

Percentage fat mass is usually used to assess cardiometabolic status, but this may be inappropriate mathematically; fat mass is the numerator, but is also included in the denominator [15]. Instead, the fat mass to fat-free mass ratio (F2FFMR) may be a superior indicator of the ability to maintain homoeostasis at the level of the organ or tissue. F2FFMR can be used for the prediction of metabolic risk in population-based studies [16]. However, it is not clear whether changes in F2FFMR have effects, and how fat mass and fat-free mass ratio changes. To date, some studies [17–19] explored the body composition changes based on small-scale survey and short-term follow-up in different groups of population, few studies have characterized the trajectories of F2FFMR and their associations with the development of cardiometabolic risk factors in adults based large-scale population-based longitudinal study in China.

We hypothesized that an accumulation of body fat over time would be associated with the development of cardiometabolic risk factors. In the present study, we aimed to evaluate the association of body composition trajectory with the prevalence of cardiometabolic risk factors (dyslipidemia, diabetes, and hypertension), to better inform obesity control and prevention.

## 2. Methods

### 2.1. Study design

We used data from the China Health and Nutrition Survey (CHNS) to characterize body composition trajectory. Data were accessed from the Carolina Population Center (http://www.cpc.unc.edu/projects/china). The CHNS is an ongoing, large-scale, open, longitudinal, household-based survey that is conducted in China [20]. Nine provinces were selected, and a multi-stage random cluster sampling method stratified by income was used in each province. The first wave of the CHNS was completed in 1993, which was followed by subsequent waves in 1997, 2000, 2004, 2006, and 2009. A detailed description of the survey has been published elsewhere [20]. This study was approved by the Institutional Review Board of the National Institute for Nutrition and Food Safety, China Center for Disease Control and Prevention, and the University of North Carolina at Chapel Hill. All the participants provided their written informed consent.

### 2.2. Study population

The study cohort comprised adults aged 18–60 years at baseline in 1993 for whom age, sex, and physical examination data [skinfold thickness, body mass, height, systolic blood pressure (SBP), and diastolic blood pressure (DBP)] were available. Participants who were pregnant at the time of the survey, for whom data were missing or biologically implausible, or who had cancer were excluded. Those for whom data from at least two rounds of the survey were available were included. Ultimately, 3,013 participants were studied, of whom 1,637 were women. A flow chart for the study is shown in Fig 1.

### 2.3. Measurement and definition of body fat mass

Skinfold thickness, height, and body mass were measured using standard protocols [21]. Skinfold thickness was measured three times using skinfold calipers over the triceps muscle on the

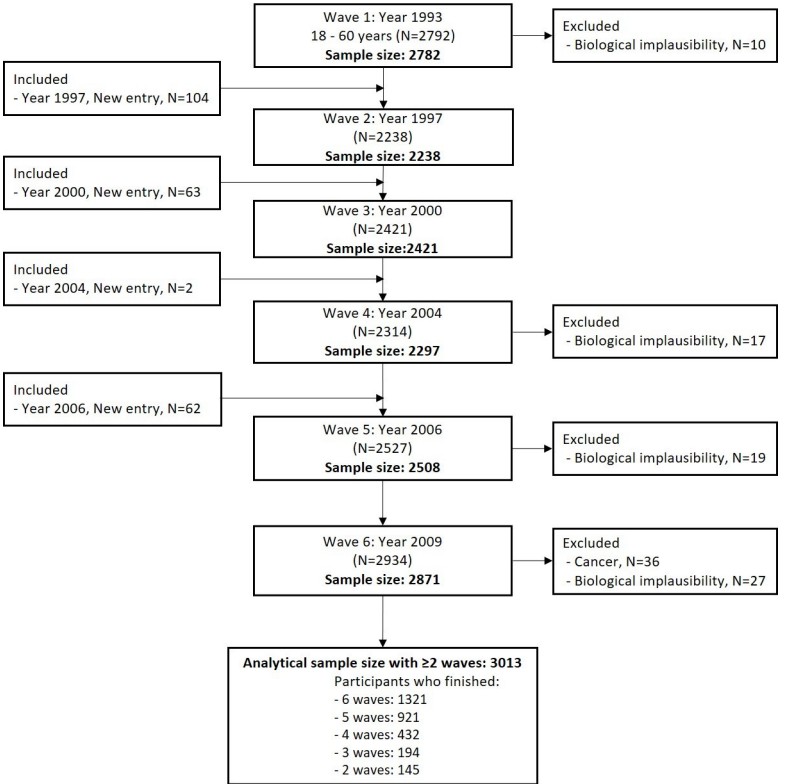

**Fig 1. Flow chart of the derivation of the study cohort.**

right arm, and the mean value was used in analyses. Height was measured without shoes using a Seca stadiometer (Seca North America East, Hanover, MD, USA), and body mass was measured while wearing lightweight clothing using a calibrated beam balance. BMI was calculated by dividing body mass (in kilograms) by the square of height (in meters). Weight status was defined as BMI $\geq$24 kg/m$^2$ for overweight and $\geq$28 kg/m$^2$ for obesity, respectively. Body fat mass was estimated using equations that included BMI and skinfold thickness: for men, fat mass percentage (FMP) = (0.742 × BMI) + (0.950 × triceps skinfold) + (0.335 × age) − 20.0; and for women, FMP (%) = (0.730 × BMI) + (0.548 × triceps skinfold) + (0.270 × age) − 5.5 [22]. Fat mass was calculated as FMP × body mass, and fat-free mass was calculated as body mass − fat mass. F2FFMR was calculated by dividing fat mass by fat-free mass.

## 2.4. Cardiometabolic risk factors

SBP and DBP were measured three times using the right arm after 10 min of rest in a seated position, using mercury sphygmomanometers with appropriate cuff sizes [23], and the mean values were used in analyses. Hypertension was defined as SBP/DBP $\geq$140/90 mm Hg, or the use of antihypertensive drugs, or a self-reported diagnosis of hypertension [23, 24]. After an overnight fast, blood sample was collected and biochemical test was completed in 2009. Total cholesterol (TC), triglyceride (TG), and high- and low- density lipoprotein–cholesterol (HDL-C and LDL-C) were measured using the glycerol-phosphate oxidase method on a Hitachi 7600 automated analyzer (Tokyo, Japan). High TC was defined as $\geq$6.2 mmol/l, high TG as $\geq$2.3 mmol/l, high LDL-C as $\geq$4.1 mmol/l, and low HDL-C as <1.0 mmol/l. Dyslipidemia was defined as any of high TC, high TG, high LDL-C, or low HDL-C, according to the

guidelines for the prevention and treatment of dyslipidemia in Chinese adults [25]. Fasting plasma glucose (FPG) was measured using the glucose oxidase-phenol and 4-aminophenazone method (Randox Laboratories Ltd., Crumlin, UK), and diabetes was defined as FPG ≥7.0 mmol/l or the use of anti-diabetic medication.

### 2.5. Covariates and definitions

Educational level was classified according to attendance at junior high school or below, high or technical school, or college and above. Physical activity was categorized as light, moderate, or heavy. Income was categorized according to individual net income per year as <$3,000, $3,000–$4,999 and ≥$5,000. Marital status was defined as married or single. Living conditions were defined as urban or rural. Smoking was defined as the use of any nicotine-based product in the preceding year, and the participants were classified as smokers, non-smokers, or former smokers. Alcohol consumption was defined using the previous year's consumption, and participants were classified as drinkers, non-drinkers, and former drinkers. Age and weight status were also adjusted in the model simultaneously.

### 2.6. Statistical analysis

Body fat trajectory patterns were identified using the group-based trajectory modeling method [26] and longitudinal body fat data. Model fitting and parameter estimation were performed using the maximum likelihood method. Specific trajectory patterns were identified using the Bayes information criterion (BIC) in the group-based trajectory modeling [26, 27]. The most appropriate models were considered to be those that permitted the most homogeneous grouping of the individual patterns, selected from among those with low BIC values. The minimum sample size for each trajectory group was 3% of the total cohort. For both men and women, a quadratic model was selected, and four and three trajectory groups were identified, respectively (Table 1). According to the trends in each trajectory group between 1993 and 2009, they were labelled "Low stability", "High stability", "Increase and decrease", and "Increasing" in men; and "Low stability", "High stability", and "Increasing" in women (Fig 2).

In the "Low stability" groups, F2FFMR remained low and did not vary substantially. In the "High stability" groups, F2FFMR remained high and did not vary substantially. In the "Increase and decrease" group, F2FFMR started low, then increased, before decreasing to a low level again. In the "Increasing" groups, F2FFMR increased, from low to high. Each participant was assigned to one of these groups and their basic characteristics were compared using the chi-square test for categorical variables and the ANOVA F test for continuous variables. Multinomial logistic regression was used to assess the relationships between the body composition trajectory group and cardiometabolic risk factors, with the Low stability group as the reference. All the analyses were stratified according to sex. In addition, all the socioeconomic, demographic, and lifestyle covariates were included in the final multivariate analysis model. Age and weight status at the final visit were included as covariates in the adjusted analysis. SAS 9.4 (Cary, NC, USA) was used for the data analysis and trajectory analysis was performed using Mplus 8.3 software (Los Angeles, CA, USA). All the statistical tests were two-sided, and $P \leq 0.05$ was regarded as indicating statistical significance.

## 3. Results

A total of 3,013 individuals (1,637 women and 1,376 men) were included in the study. The men were allocated to the following groups: 83.9% Low stability, 3.6% High stability, 6.8% Increase and decrease, and 5.7% Increasing. There were no differences in marital status or alcohol consumption among these groups, but there were differences in age, location,

**Table 1. Indices of goodness-of-fit for the latent class growth analysis.**

| Sex | Model | Model fit | 2 groups | 3 groups | 4 groups | 5 groups |
|---|---|---|---|---|---|---|
| Male | Linear | BIC | -7477.424 | -7586.715 | -7650.447 | -7663.206 |
| | | Entropy | 0.952 | 0.864 | 0.866 | 0.861 |
| | | LMR P-value | 0.0095 | 0.0015 | 0.0032 | 0.4822 |
| | | BLRT P-value | 0 | 1 | 1 | 1 |
| | | Smallest Prop. | 3.50% | 3.40% | 1.20% | 1.20% |
| | Quadratic | BIC | -7602.546 | -7767.145 | **-7900.029** | -7880.402 |
| | | Entropy | 0.963 | 0.895 | **0.88** | 0.893 |
| | | LMR P-value | 0.0029 | 0.0034 | **0.0069** | 0.0351 |
| | | BLRT P-value | 0 | 0 | **1** | 1 |
| | | Smallest Prop. | 3.70% | 3.80% | **3.60%** | 0.40% |
| Female | Linear | BIC | -14296.499 | -14396.825 | -14407.113 | -14418.243 |
| | | Entropy | 0.849 | 0.869 | 0.871 | 0.806 |
| | | LMR P-value | 0 | 0 | 0.0043 | 0.0072 |
| | | BLRT P-value | 0 | 0 | 1 | 1 |
| | | Smallest Prop. | 8.50% | 3.70% | 1.60% | 1.20% |
| | Quadratic | BIC | -14551.652 | **-14632.521** | -14656.953 | -14706.93 |
| | | Entropy | 0.869 | **0.875** | 0.875 | 0.87 |
| | | LMR P-value | 0.2383 | **0.2348** | 0.232 | 0.2323 |
| | | BLRT P-value | 0 | **0.995** | 1 | 0.985 |
| | | Smallest Prop. | 7.90% | **3.50%** | 0.70% | 0.70% |

educational level, individual net income, smoking, physical activity, body mass, and FMP. The women were allocated to the following groups: 88.9% Low stability, 7.6% High stability, and 3.5% Increasing. There were no differences in educational level, individual net income, or alcohol consumption among these groups, but there were differences in the other factors (Tables 2 and 3).

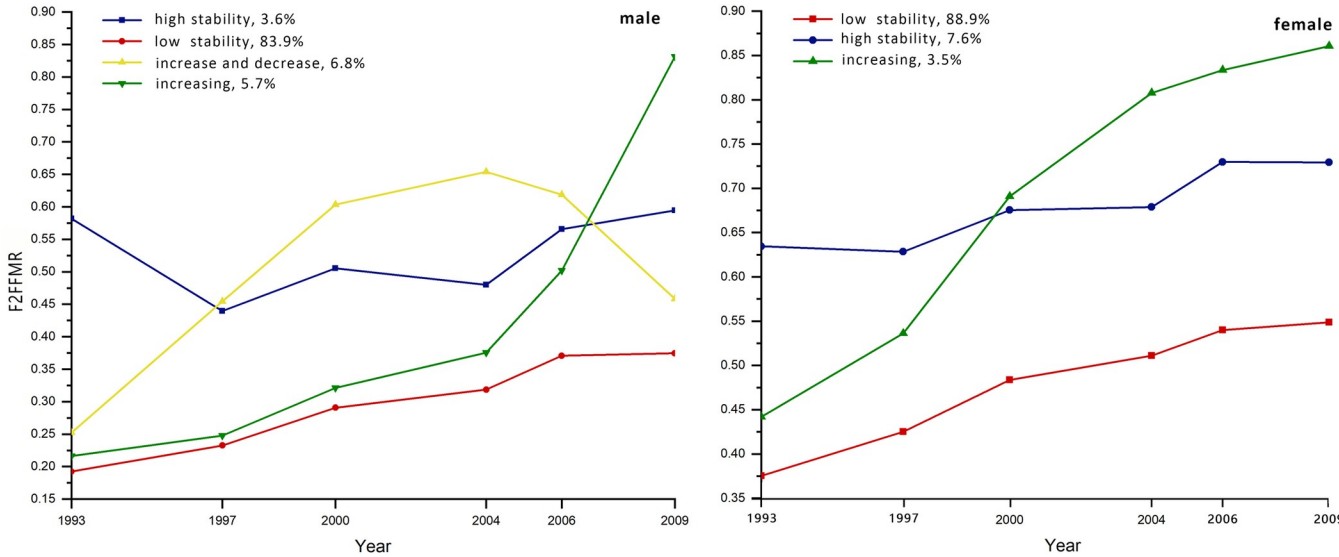

**Fig 2. Latent body fat trajectories of each group between 1993 and 2009, identified using group-based trajectory modeling.**

**Table 2. Characteristics of the male participants, according to their latent F2FFMR trajectory group.**

| Characteristics n (%)/mean (SD) | F2FFMR trajectory group (n = 1376) | | | | F/χ2 | P value |
|---|---|---|---|---|---|---|
| | Low stability (n = 1155) | High stability (n = 49) | Increase and decrease (n = 94) | Increasing (n = 78) | | |
| Age (years), mean (SD) | | | | | | |
| 1993 | 37.8(10.5) | 45.1(9.2) | 43.0(9.1) | 39.5(10.6) | 14.6 | <0.001 |
| 2009 | 53.7(10.5) | 61.1(9.2) | 59.0(9.1) | 55.5(10.6) | 14.8 | <0.001 |
| Married | 1071(92.7) | 44(89.8) | 90(95.7) | 71(91) | 2.23 | 0.526 |
| Location | | | | | | |
| Rural | 898(77.8) | 29(59.2) | 69(73.4) | 62(79.5) | 10.064 | 0.018 |
| Urban | 257(22.2) | 20(40.8) | 25(26.6) | 16(20.5) | | |
| Education | | | | | | |
| Lower than junior high school | 941(81.5) | 34(69.4) | 71(75.5) | 58(74.4) | 43.5 | <0.001 |
| High or technical school | 145(12.6) | 1(2.0) | 11(11.7) | 11(14.1) | | |
| College and above | 69(6.0) | 14(28.6) | 12(12.8) | 9(11.5) | | |
| Individual net income per year | | | | | | |
| <$3000 | 721(80.6) | 22(57.9) | 59(73.8) | 51(79.7) | 13.7 | 0.033 |
| $3000–$4999 | 109(12.2) | 10(26.3) | 15(18.8) | 8(12.5) | | |
| ≥$5000 | 65(7.3) | 6(15.8) | 6(7.5) | 5(7.8) | | |
| Smoking | | | | | | |
| No | 59(5.8) | 1(2.4) | 8(9.9) | 5(6.9) | 14.4 | 0.026 |
| Yes | 745(72.9) | 27(65.9) | 45(55.6) | 48(66.7) | | |
| Quit | 218(21.3) | 13(31.7) | 28(34.6) | 19(26.4) | | |
| Alcohol consumption | | | | | | |
| No | 101(8.8) | 3(6.1) | 6(6.4) | 8(10.3) | 3.91 | 0.688 |
| Yes | 708(61.4) | 30(61.2) | 52(55.3) | 48(61.5) | | |
| Quit | 345(29.9) | 16(32.7) | 36(38.3) | 22(28.2) | | |
| Physical activity | | | | | | |
| Light | 408(35.7) | 26(54.2) | 44(48.9) | 32(42.1) | 17.897 | 0.006 |
| Moderate | 171(15.0) | 9(18.8) | 14(15.6) | 7(9.2) | | |
| Heavy | 563(49.3) | 13(27.1) | 32(35.6) | 37(48.7) | | |
| Weight status 2009 | | | | | | |
| BMI <24 | 736(66.4) | 10(21.3) | 29(32.2) | 31(40.3) | 134 | <0.001 |
| BMI≥24, <28 | 328(29.6) | 28(59.6) | 39(43.3) | 32(41.6) | | |
| BMI ≥28 | 44(4.0) | 9(19.2) | 22(24.4) | 14(18.2) | | |
| FMP, mean (SD) | | | | | | |
| 1993 | 15.9(6.6) | 36.5(4.8) | 20.9(10.2) | 17.6(7.5) | 142.97 | <0.001 |
| 2009 | 26.9(7.9) | 38.1(10.0) | 31.2(7.8) | 45.8(4.5) | 167.26 | <0.001 |

In logistic regression models 1–4, cardiometabolic risk factors were used as dependent variables and trajectory group was the independent variable. The Low stability group was the reference group in each model. In men, high TC and high LDL-C were not associated with F2FFMR trajectory group, regardless of whether they were adjusted for covariates or not. Prior to adjustment for any covariates, the Increase and decrease trajectory group was significantly associated with high TG [crude odds ratio (OR) 1.65, 95% confidence interval (CI) 1.00–2.70]. When adjusted for age in 2009, the Increase and decrease trajectory group remained significantly associated with high TG (crude OR 1.87, 95% CI 1.13–3.10). However, when further adjusted for body mass in 2009, this association disappeared. The High stability group was significantly associated with low HDL-C (crude OR 2.06, 95% CI 1.00–4.24) prior to

**Table 3. Characteristics of the female participants, according to their latent F2FFMR trajectory group.**

| Characteristics n (%)/mean (SD) | F2FFMr trajectory group (n = 1637) | | | F/χ2 | P value |
|---|---|---|---|---|---|
| | Low stability (n = 1456) | High stability (n = 124) | Increasing (n = 57) | | |
| Age (years), mean (SD) | | | | | |
| 1993 | 38.9(9.6) | 47.1(7.9) | 41.9(8.1) | 44.1 | <0.001 |
| 2009 | 54.9(9.6) | 63.0(7.9) | 57.8(8.1) | 43.8 | <0.001 |
| Married | 1275(87.6) | 98(79.0) | 48(84.2) | 7.617 | 0.022 |
| Location | | | | | |
| Rural | 1120(76.9) | 83(66.9) | 47(82.5) | 7.530 | 0.023 |
| Urban | 336(23.1) | 41(33.1) | 10(17.5) | | |
| Education | | | | | |
| Lower than junior high school | 1314(90.3) | 112(90.3) | 53(93.0) | 9.379 | 0.052 |
| High or technical school | 104(7.1) | 4(3.2) | 2(3.5) | | |
| College and above | 38(2.6) | 8(6.5) | 2(3.5) | | |
| Individual net income per year | | | | | |
| < $3000 | 899(88.6) | 67(87.0) | 29(80.6) | 2.669 | 0.615 |
| $3000 –$4999 | 74(7.3) | 7(9.1) | 4(11.1) | | |
| ≥$5000 | 42(4.1) | 3(3.9) | 3(8.3) | | |
| Smoking | | | | | |
| No | 744(89.9) | 49(73.1) | 17(60.7) | 37.089 | <0.001 |
| Yes | 37(4.5) | 8(11.9) | 6(21.4) | | |
| Quit | 47(5.7) | 10(14.9) | 5(17.9) | | |
| Alcohol consumption | | | | | |
| No | 1029(71.1) | 83(69.2) | 42(73.7) | 1.977 | 0.74 |
| Yes | 119(8.2) | 14(11.7) | 4(7.0) | | |
| Quit | 299(20.7) | 23(19.2) | 11(19.3) | | |
| Physical activity | | | | | |
| Light | 664(46.3) | 89(71.8) | 34(59.7) | 33.318 | <0.001 |
| Moderate | 162(11.3) | 8(6.5) | 7(12.3) | | |
| Heavy | 607(42.4) | 27(21.8) | 16(28.1) | | |
| Weight Status 2009 | | | | | |
| BMI <24 | 865(61.4) | 28(23.5) | 0 | 285.679 | <0.001 |
| BMI≥24, <28 | 440(31.2) | 48(40.3) | 18(33.3) | | |
| BMI≥28 | 105(7.5) | 43(36.1) | 36(66.7) | | |
| FMP, mean (SD) | | | | | |
| 1993 | 27.2(5.0) | 39.5(3.1) | 30.5(4.6) | 363.8 | <0.001 |
| 2009 | 35.2(5.3) | 42.2(5.5) | 47.9(4.8) | 236.4 | <0.001 |

adjustment, and after adjustment for age in 2009, the adjusted OR and 95% CI were 2.37 (1.13–4.94). However, after further adjustment for body mass status in 2009, this association disappeared.

The Increase and decrease group was significantly associated with dyslipidemia prior to adjustment (crude OR, 95% CI: 1.64, 1.07–2.53), and after adjustment for age in 2009 (adjusted OR, 95% CI: 1.79, 1.16–2.76). In addition, the High stability group was significantly associated with dyslipidemia after adjustment for age in 2009 (adjusted OR, 95% CI: 1.95, 1.08–3.51). However, after further adjustment for body mass in 2009, this association disappeared. The High stability group and the Increase and decrease group were significantly associated with diabetes after adjustment for age and body mass in 2009 (adjusted OR, 95% CI: 2.68, 1.31–5.51

and 1.90, 1.04–3.46, respectively). After further adjustment for educational level, smoking, alcohol consumption, location, marital status, and physical activity in model 4, the High stability group remained significantly associated (adjusted OR, 95% CI: 2.72, 1.25–5.92), but the association disappeared in the Increase and decrease group. After adjustment for age in 2009, the High stability group and the Increase and decrease group were also significantly associated with hypertension (adjusted OR, 95% CI: 2.48, 1.32–4.68 and 1.75, 1.12–2.72, respectively), but after further adjustment for other covariates, this association disappeared (Table 4).

**Table 4. ORs and 95% CIs for the associations between the F2FFMR trajectories of men and cardiometabolic risk factors in 2009.**

| Cardiometabolic risk factors | F2FFMR trajectory group | Model 1,OR(95%CI) | Model 2,OR(95%CI) | Model 3,OR(95%CI) | Model 4,OR(95%CI) |
|---|---|---|---|---|---|
| High TC | | | | | |
| | Low stability | Reference | Reference | Reference | Reference |
| | High stability | 0.53(0.13,2.21) | 0.54(0.12,2.27) | 0.45(0.11,1.95) | 0.23(0.03,1.73) |
| | Increase and decrease | 1.00(0.45,2.23) | 1.01(0.45,2.27) | 0.87(0.38,2.00) | 0.82(0.35,1.93) |
| | Increasing | 1.42(0.66,3.05) | 1.43(0.66,3.06) | 1.24(0.57,2.70) | 1.06(0.46,2.42) |
| High TG | | | | | |
| | Low stability | Reference | Reference | Reference | Reference |
| | High stability | 1.65(0.84,3.21) | 1.96(0.99,3.88) | 1.11(0.54,2.29) | 0.82(0.36,1.85) |
| | Increase and decrease | **1.65(1.00,2.70)***  | **1.87(1.13,3.10)***  | 1.09(0.63,1.88) | 1.06(0.59,1.88) |
| | Increasing | 0.58(0.27,1.22) | 0.60(0.28,1.28) | 0.36(0.16,0.78) | 0.32(0.14,0.74) |
| High LDL-C | | | | | |
| | Low stability | Reference | Reference | Reference | Reference |
| | High stability | 0.65(0.20,2.12) | 0.66(0.20,2.17) | 0.52(0.16,1.76) | 0.50(0.14,1.71) |
| | Increase and decrease | 0.80(0.36,1.78) | 0.81(0.36,1.80) | 0.65(0.29,1.50) | 0.54(0.22,1.31) |
| | Increasing | 1.58(0.81,3.09) | 1.59(0.81,3.11) | 1.37(0.69,2.72) | 1.24(0.60,2.550 |
| Low HDL-C | | | | | |
| | Low stability | Reference | Reference | Reference | Reference |
| | High stability | **2.06(1.00,4.24)***  | **2.37(1.13,4.94)***  | 1.37(0.63,2.96) | 1.66(0.73,3.74) |
| | Increase and decrease | 1.17(0.62,2.21) | 1.29(0.68,2.44) | 0.74(0.38,1.47) | 0.69(0.33,1.43) |
| | Increasing | 1.00(0.49,2.07) | 1.04(0.51,2.14) | 0.69(0.32,1.45) | 0.57(0.24,1.35) |
| Dyslipidemia | | | | | |
| | Low stability | Reference | Reference | Reference | Reference |
| | High stability | 1.74(0.97,3.10) | **1.95(1.08,3.51)***  | 1.17(0.62,2.19) | 1.01(0.51,2.01) |
| | Increase and decrease | **1.64(1.07,2.53)***  | **1.79(1.16,2.76)***  | 1.12(0.69,1.79) | 1.07(0.64,1.77) |
| | Increasing | 1.09(0.67,1.79) | 1.12(0.69,1.84) | 0.73(0.43,1.24) | 0.66(0.38,1.16) |
| Diabetes | | | | | |
| | Low stability | Reference | Reference | Reference | Reference |
| | High stability | **4.33(2.21,8.45)***  | **3.70(1.87,7.32)***  | **2.68(1.31,5.51)***  | **2.72(1.25,5.92)***  |
| | Increase and decrease | **3.03(1.75,5.25)***  | **2.72(1.56,4.74)***  | **1.90(1.04,3.46)***  | 1.85(0.98,3.49) |
| | Increasing | 0.48(0.15,1.55) | 0.46(0.14,1.48) | 0.35(0.11,1.16) | 0.39(0.12,1.28) |
| hypertension | | | | | |
| | Low stability | Reference | Reference | Reference | Reference |
| | High stability | **3.40(1.84,6.30)***  | **2.48(1.32,4.68)***  | 1.53(0.79,2.96) | 1.55(0.76,3.16) |
| | Increase and decrease | **2.14(1.39,3.30)***  | **1.75(1.12,2.72)***  | 1.10(0.69,1.77) | 1.06(0.64,1.76) |
| | Increasing | 1.29(0.81,2.05) | 1.20(0.74,1.94) | 0.84(0.51,1.39) | 0.92(0.55,1.55) |

*P<0.05. Model 1: unadjusted; Model 2: adjusted for age in 2009; Model 3: Model 2 + body mass in 2009; Model 4: Model 3 + smoking, drinking, physical activity, location, educational level, and marital status.

**Table 5. ORs and 95% CIs for the associations between the F2FFMR trajectories of women and cardiometabolic risk factors in 2009.**

| Cardiometabolic risk factors | F2FFMR trajectory group | Model 1,OR(95%CI) | Model 2,OR(95%CI) | Model 3,OR(95%CI) | Model 4,OR(95%CI) |
|---|---|---|---|---|---|
| High TC | | | | | |
| | Low stability | Reference | Reference | Reference | Reference |
| | High stability | **2.25(1.44,3.51)*** | **1.62(1.02,2.58)*** | 1.38(0.84,2.25) | 1.68(0.88,3.23) |
| | Increasing | 1.38(0.67,2.86) | 1.25(0.60,2.60) | 0.94(0.43,2.07) | 0.96(.034,2.72) |
| High TG | | | | | |
| | Low stability | Reference | Reference | Reference | Reference |
| | High stability | **2.61(1.75,3.89)*** | **2.41(1.60,3.65)*** | 1.35(0.86,2.13) | 1.23(0.66,2.27) |
| | Increasing | **2.64(1.49,4.66)*** | **2.57(1.45,4.54)*** | 1.00(0.53,1.86) | 1.02(.043,2.42) |
| High LDL-C | | | | | |
| | Low stability | Reference | Reference | Reference | Reference |
| | High stability | **2.15(1.39,3.30)*** | **1.60(1.03,2.50)*** | 1.37(0.86,2.20) | 1.59(0.85,2.99) |
| | Increasing | 1.33(0.66,2.68) | 1.20(0.59,2.45) | 0.93(0.44,1.97) | 0.94(0.34,2.62) |
| Low HDL-C | | | | | |
| | Low stability | Reference | Reference | Reference | Reference |
| | High stability | 1.72(0.93,3.18) | **1.92(1.01,3.63)*** | 1.27(0.75,2.50) | 1.29(0.52,3.21) |
| | Increasing | 1.13(0.40,3.18) | 1.17(0.41,3.32) | 0.61(0.20,1.81) | 0.22(0.03,1.77) |
| Dyslipidemia | | | | | |
| | Low stability | Reference | Reference | Reference | Reference |
| | High stability | **2.63(1.81,3.81)*** | **2.22(1.52,3.25)*** | **1.55(1.03,2.34)*** | 1.47(0.84,2.56) |
| | Increasing | 1.63(0.95,2.78) | 1.54(0.89,2.64) | 0.79(0.44,1.43) | 0.91(040,2.09) |
| Diabetes | | | | | |
| | Low stability | Reference | Reference | Reference | Reference |
| | High stability | **4.52(2.89,7.08)*** | **3.59(2.25,5.73)*** | **3.17(1.91,5.27)*** | **3.06(1.54,6.08)*** |
| | Increasing | **2.44(1.16,5.11)*** | **2.27(1.08,4.76)*** | 1.79(0.80,4.05) | 1.81(0.59,5.61) |
| hypertension | | | | | |
| | Low stability | Reference | Reference | Reference | Reference |
| | High stability | **4.03(2.72,5.98)*** | **2.58(1.71,3.90)*** | **1.63(1.05,2.53)*** | **2.05(1.09,3.85)*** |
| | Increasing | **2.89(1.66,5.04)*** | **2.62(1.47,4.65)*** | 1.19(0.64,2.21) | 1.13(0.47,2.76) |

*P<0.05. Model 1: unadjusted; Model 2: adjusted for age in 2009; Model 3: Model 2 + body mass in 2009; Model 4: Model 3 + smoking, drinking, physical activity, location, educational level, and marital status.

In women, prior to adjustment, the High stability group was associated with high TC (crude OR, 95% CI: 2.25, 1.44–3.51). The High stability and Increasing groups were significantly associated with high TG, with crude ORs (95% CIs) of 2.61 (1.75–3.89) and 2.64 (1.49–4.66), respectively. The High stability group was associated with high LDL-C (crude OR, 95% CI: 2.15, 1.39–3.30) and dyslipidemia (crude OR, 95% CI: 2.63, 1.81–3.81). In addition, the High stability and Increasing groups were significantly associated with diabetes (crude OR, 95% CI: 4.52, 2.89–7.08 and 2.44, 1.16–5.11, respectively) and hypertension (crude OR, 95% CI: 4.03, 2.72–5.98 and 2.89, 1.66–5.04, respectively) (Table 5).

In women, after adjustment for age in 2009, the same associations were identified (Table 5). After further adjustment for body mass in 2009 in model 3, the associations with high TC, high TG, high LDL-C, and low HDL-C disappeared. There was a significant association between the High stability group and dyslipidemia (crude OR, 95% CI: 1.55, 1.03–2.34), but this association disappeared after adjustment for other covariates in model 4. The High stability group was also significantly associated with diabetes and hypertension (adjusted OR, 95% CI: 3.06, 1.54–6.08 and 2.05, 1.09–3.85, respectively) (Table 5).

## 4. Discussion

In the present study, we have identified four patterns of F2FFMR trajectory in men and three in women using data from a 16-year population-based cohort study. By comparing the risks of dyslipidemia, diabetes, and hypertension among the participants with these different patterns, we determined that the participants in the High stability group were at the highest risks of diabetes and hypertension. Through comparing impact of the high stability group, increasing group, increase and decrease group on diabetes. We speculate the effect of F2FFMR on cardiometabolic risk is likely to accumulate slowly, and high F2FFMR in early life might have an impact on diabetes in later life. Our findings show that the monitoring of F2MMR trajectory may help identify individuals who are at higher risk of diabetes and hypertension.

In the present study, we found dyslipidemia was not statistically associated with F2MMR trajectory after adjustment for age and weight status in men, lifestyle covariates in women. So dyslipidemia might be primarily determined by body mass in men, but by lifestyle in women. The body composition of men and women differs: adipose tissue is more likely to accumulate around the trunk and abdomen of men, but around the hips and thighs of women [28, 29]. Study of the F2FFMR indicates that the adverse effects of high fat mass can be offset by the protective effects of high fat-free mass. Specifically, high fat-free mass protects against high TC and high LDL-C [30]. However, F2FFMR reflects whole-body composition, and does not discriminate between the effects of body fat in differing locations. For example, excess accumulation of fat mass, especially in the upper body, is associated with dyslipidemia in normal-weight individuals [31]. In addition, visceral fat is a risk factor for dyslipidemia in men, and this effect is independent of the influence of BMI and waist circumference [32], but might not be a risk factor in women [33]. BMI is an independent risk factor for hypertension in both men and women, and high fat mass is associated with a higher risk of hypertension, even in non-obese populations [34]. In contrast, it has also been shown that a reduction in fat-free mass is more strongly associated with the normalization of blood pressure than a reduction in fat mass [35].

The present findings also suggest that the impact of F2FFMR trajectory on cardiometabolic outcomes is mediated by current body mass. A study of a cohort from birth has also shown that the trajectory of fat mass is associated with the development of cardiometabolic risk factors in adulthood and is affected by BMI in adulthood [36]. In both men and women, F2FFMR of Hight stability was associated with diabetes in the present study, and this association remained even after adjustment for the covariates and body mass. The fat-to-muscle ratio is associated with blood glucose [37] and visceral fat mass might predict the risk of prediabetes or diabetes [38]. Fat-free mass, which mainly consists of muscle, is protective against diabetes, and skeletal muscle plays an important role in the consumption and storage of glucose [39], the regulation of blood glucose, and the prevention of hyperglycemia [40]. Thus, people with low muscle mass are at a higher risk of developing type 2 diabetes than those with high muscle mass, and the lower the percentage muscle mass, the higher the risk of developing type 2 diabetes [41, 42]. Furthermore, visceral fat has an independent effect on cardiometabolic risk factors, such as abnormal lipid and glucose metabolism [30, 37]. However, it was not possible to analyze the effect of visceral fat mass alone in the present study.

In the present study we used data from the CHNS, which is a nationwide study that has been conducted for over 16 years. Therefore, this study is meaningful because it is the first study to use group-based trajectory modeling method to analyze the change of F2FFMR in Chinese, and the results provide strong evidence for associations between long-term changes in body composition and the development of cardiometabolic risk factors. We first use the F2FFMR as a metabolic load-capacity indicator for study with the cardiometabolic outcome. The latent growth model was used to create the body fat trajectory groups. However, there

were some limitations to the study. First, the fat mass and fat-free mass were estimated using a verified model that included BMI and upper arm skinfold thickness, rather than by direct measurement. Therefore, there may be some bias in the body composition data. Second, the trajectory groups, except for the Low stability group, were relatively small. Larger samples are required to verify the association of F2FFMR trajectory with dyslipidemia and other cardiometabolic risk factors. Third, blood pressure was measured once, whereas a clinical diagnosis of hypertension should be made on the basis of three measurements made on three different days. Due to unavailability of data, we could not distinguish between type 1 diabetes and type 2 diabetes in this study. Fourth, F2FFMR trajectory might be affected by differences in ethnicity and eating habits. The present study was of the Chinese adult population; therefore, the findings require confirmation in other populations.

In conclusion, four types of F2FFMR trajectory were identified in men and three in women. High stability trajectory of F2FFMR was associated with the highest risk of developing diabetes in men, and diabetes and hypertension in women, independent of age and current body mass. Our results also suggest that the association between F2FFMR trajectory and dyslipidemia and hypertension in men is mediated by current body mass.

## Acknowledgments

This research used data from CHNS. We thank the National Institute for Nutrition and Health, China Center for Disease Control and Prevention, Carolina Population Center (P2C HD050924, T32 HD007168), the University of North Carolina at Chapel Hill, the NIH (R01-HD30880, DK056350, R24 HD050924, and R01-HD38700) and the NIH Fogarty International Center (D43 TW009077, D43 TW007709) for support for the CHNS data collection and analysis of files from 1989 to 2015 and future surveys, the China–Japan Friendship Hospital, the Ministry of Health for support for CHNS 2009, the Chinese National Human Genome Center in Shanghai since 2009, and the Beijing Municipal Center for Disease Prevention and Control since 2011. We also thank Mark Cleasby, PhD, from Edanz Group (https://en-author-services.edanz.com/) for editing a draft of this manuscript.

## Author Contributions

**Conceptualization:** Zhaoyang Fan, Junting Liu.

**Data curation:** Zhaoyang Fan, Yunping Shi.

**Formal analysis:** Zhaoyang Fan, Yunping Shi, Guimin Huang, Dongqing Hou, Junting Liu.

**Funding acquisition:** Zhaoyang Fan, Junting Liu.

**Methodology:** Zhaoyang Fan, Junting Liu.

**Writing – original draft:** Zhaoyang Fan, Junting Liu.

**Writing – review & editing:** Zhaoyang Fan, Junting Liu.

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
