## [Decision Letter · Decision Letter 0]

29 Jan 2021

PONE-D-20-38516

Body Composition Trajectory and Association with Cardiometabolic Risk Factors: a Population-Based Cohort Study

PLOS ONE

Dear Dr. Liu,

Thank you for submitting your manuscript to PLOS ONE. After careful consideration, we feel that it has merit but does not fully meet PLOS ONE’s publication criteria as it currently stands. Therefore, we invite you to submit a revised version of the manuscript that addresses the points raised during the review process.

We look forward to receiving your revised manuscript.

Kind regards,

Yiqiang Zhan

Academic Editor

PLOS ONE

Journal Requirements:

Reviewers' comments:

Reviewer's Responses to Questions

**Comments to the Author**

1. Is the manuscript technically sound, and do the data support the conclusions?

Reviewer #1: Yes

Reviewer #2: No

2. Has the statistical analysis been performed appropriately and rigorously? 

Reviewer #1: Yes

Reviewer #2: I Don't Know

3. Have the authors made all data underlying the findings in their manuscript fully available?

Reviewer #1: Yes

Reviewer #2: Yes

4. Is the manuscript presented in an intelligible fashion and written in standard English?

Reviewer #1: No

Reviewer #2: No

5. Review Comments to the Author

Reviewer #1: Many thanks for the opportunity to review this manuscript. The comments below are intended as a constructive direction on improving the manuscript.

The manuscript will benefit from editing to tidy up grammar/spelling.

Reviewer #2: Overview

The authors present an interesting analysis which considers different patterns of change in fat to fat-free mass ratio (trajectories) across 16 years in a population-based cohort in China. They relate the different trajectories to cardiometabolic outcomes. I think the concept and the results are interesting however I have some major concerns about the discussion and some of the conclusions which have been drawn from the results.

Furthermore, it is sometimes difficult to follow the points being made (particularly in the introduction & discussion). It is essential that the whole manuscript undergoes a thorough proof reading to ensure that all statements are written correctly in English and make sense as they are written.

Introduction

Where the authors refer to ‘cardiovascular risk factors’, this includes dyslipidemia, diabetes, cholesterol and triglyceride measures; I would consider these as ‘cardiometabolic’ outcomes and suggest changing the general terminology throughout the manuscript.

It would be good to define the ‘trajectory effect’ more clearly and mention potential biological underpinning for the trajectory effect itself, not just the negative consequences of obesity.

F2MR, consider changing this to F2FFMR

Methods

The first paragraph states that the first wave was 1989, then the ‘study population’ paragraph says baseline was 1993- this is a bit confusing, when was baseline?

3013 were included- how many were originally seen? Or, put another way, how many were excluded? In the flow chart it looks like 2792 participants were seen in 1993, I’m not sure I understand how 3013 participants can be included in this analysis?

Please insert the reference for ‘guidelines for the prevention and treatment of dyslipidemia in Chinese adults’.

Was there any distinction between type-2 and type-1 diabetes?

Note to editor: The group‐based trajectory modelling method was applied to the longitudinal body fat data using maximum likelihood method for model fitting and parameter estimation. I believe these are sound methods, however, I am not familiar with using them in my own work and therefore cannot fully confirm if this is appropriate.

The following sentence is unclear: ‘According to the trends of each trajectory group, low stability, high stability, increase and decrease, and increasing were designated in men. In women, low stability, high stability and increasing groups were designated (Figure 2).’

In the text it could be clearer what each group means, i.e. ‘high stability’ means a high Fat to fat-free mass ratio without much variation between 1993 and 2009. Please clearly describe why fat to fat-free mass ratio was used and not just the fat % value?

Results

I’m concerned about the high number of individuals who fall into the ‘high stability’ group. I am not a statistician so I cannot say if this invalidates the statistical methods in any way. It would be worth having the analysis verified by a statistician.

What is the justification for the adjustment strategy in tables 4 and 5? Why not have an age adjusted model before adjustment for other confounders (education, marital status etc..)? For a better understanding of mediation by age or original weight status, these factors should be sequentially adjusted for in separate models.

In the conclusion the authors state ‘Our results also suggest that the impact of the adiposity trajectory on cardiometabolic profile is mediated by concurrent weight status.’ Is this a reference to tables 4 & 5 model 3? Could this association not also be mediated by age as age is included in model 3 along with weight status?

Figures 1 & 2: I can’t see the figure legends anywhere in the submitted manuscript? These should be provided.

Tables 2 & 3: the formatting is not quite right and makes the heading difficult to read.

Discussion

The discussion is not well written and as such it is quite hard to follow the discussion points. The findings are discussed fairly superficially, more insight into potential physiological underpinnings for the different F2MR trajectories is necessary. For example, why might the high-stability trajectory group be associated with hypertension in men, independent of concurrent age and weight status? There is repetition of statements throughout the discussion which is unnecessary.

Consider opening the discussion with a summary paragraph of the key results and why they are important/novel.

Paragraph 2: ‘dyslipidaemia was mainly affected by concurrent weight status’ I don’t think you can draw this conclusion without separating age/weight adjustment in model 3.

The limitation section should also include the high number of participants who fall into the low stability group and low numbers in the other groups and discussion of potential ethnic differences in body composition trajectories.

6. PLOS authors have the option to publish the peer review history of their article (what does this mean?). If published, this will include your full peer review and any attached files.

Reviewer #1: No

Reviewer #2: No

---

## [Author Response · Author response to Decision Letter 0]

7 Mar 2021

Response to the Reviewers’ Comments

PONE-D-20-38516

Long-term Changes in Body Composition and their Relationships with Cardiometabolic Risk Factors: A Population-Based Cohort Study

（Please note that the title has been changed from Body Composition Trajectory and Association with Cardiometabolic Risk Factors: a Population-Based Cohort Study to the above one）

We are grateful for the thoughtful reviews provided by the external referees and valuable comments from the editors. We have carefully addressed each recommendation, and the manuscript is considerably stronger as a result of the modifications outlined below.

The modified text is highlighted in tracked copy in the revised manuscript.

Requests from the reviewers:

Reviewer #1: Many thanks for the opportunity to review this manuscript. The comments below are intended as a constructive direction on improving the manuscript.

1. Q: The manuscript will benefit from editing to tidy up grammar/spelling.

A: Revised as suggested

We get help from Mark Cleasby, PhD, a native English speaker, from Edanz Group (https://en-author-services.edanzgroup.com/) for editing a draft of this manuscript .

Reviewer #2: Overview

The authors present an interesting analysis which considers different patterns of change in fat to fat-free mass ratio (trajectories) across 16 years in a population-based cohort in China. They relate the different trajectories to cardiometabolic outcomes. I think the concept and the results are interesting however I have some major concerns about the discussion and some of the conclusions which have been drawn from the results.

1. Q: Furthermore, it is sometimes difficult to follow the points being made (particularly in the introduction & discussion). It is essential that the whole manuscript undergoes a thorough proof reading to ensure that all statements are written correctly in English and make sense as they are written. 

A: Revised as suggested

We get help from Mark Cleasby, PhD, a native English speaker, from Edanz Group (https://en-author-services.edanzgroup.com/) for editing a draft of this manuscript .

2. Q: Introduction

Where the authors refer to ‘cardiovascular risk factors’, this includes dyslipidemia, diabetes, cholesterol and triglyceride measures; I would consider these as ‘cardiometabolic’ outcomes and suggest changing the general terminology throughout the manuscript.

A: Revised as suggested

3. Q: It would be good to define the ‘trajectory effect’ more clearly and mention potential biological underpinning for the trajectory effect itself, not just the negative consequences of obesity.

A: Revised as suggested (Please see page 3 and 4)

4. Q: F2MR, consider changing this to F2FFMR

A: Revised as suggested

5. Q: Methods

The first paragraph states that the first wave was 1989, then the ‘study population’ paragraph says baseline was 1993- this is a bit confusing, when was baseline?

A: Revised as suggested (Please see page 4)

6. Q: 3013 were included- how many were originally seen? Or, put another way, how many were excluded? In the flow chart it looks like 2792 participants were seen in 1993, I’m not sure I understand how 3013 participants can be included in this analysis?

A: In this study, we included the participants who had at least two rounds of data. So the data was a maximum set of all 6 rounds. 

7. Q: Please insert the reference for ‘guidelines for the prevention and treatment of dyslipidemia in Chinese adults.

A: Revised as suggested (Please see page 6)

8. Q: Was there any distinction between type-2 and type-1 diabetes?

A: In this study, we could not identify type-2 or type-1 diabetes accurately, but this study was a population-based study and type-1 diabetes usually was found in children and adolescents. Our study population was mainly in adults.

9. Q: Note to editor: The group‐based trajectory modelling method was applied to the longitudinal body fat data using maximum likelihood method for model fitting and parameter estimation. I believe these are sound methods, however, I am not familiar with using them in my own work and therefore cannot fully confirm if this is appropriate.

A: The group‐based trajectory modelling method was usually used for trajectory of longitudinal quantitative data, some references were quoted in introduction of our article.

10. Q: The following sentence is unclear: ‘According to the trends of each trajectory group, low stability, high stability, increase and decrease, and increasing were designated in men. In women, low stability, high stability and increasing groups were designated (Figure 2).’In the text it could be clearer what each group means, i.e. ‘high stability’ means a high Fat to fat-free mass ratio without much variation between 1993 and 2009. 

A: Revised as suggested (Please see page 7)

11. Q: Please clearly describe why fat to fat-free mass ratio was used and not just the fat % value?

A: Revised as suggested (Please see page 4).

12. Q: Results

I’m concerned about the high number of individuals who fall into the ‘high stability’ group. I am not a statistician so I cannot say if this invalidates the statistical methods in any way. It would be worth having the analysis verified by a statistician.

A: We have asked a statistician for verification.

In table 4 and 5, the ORs of high stability are not very big and their CIs are narrow 

And an article published in ‘Journal of clinical hypertension’ used the same method and its four groups number are 1045, 327,75 and 37, which is less than this study. Its sensitivity analysis was through Logistic regression.

Fan et, al. J Clin Hypertens. 2020;00:1–6. DOI: 10.1111/jch.14001 

13. Q: What is the justification for the adjustment strategy in tables 4 and 5? Why not have an age adjusted model before adjustment for other confounders (education, marital status etc..)? 

A: Revised as suggested (Please see page 4).

14. Q: For a better understanding of mediation by age or original weight status, these factors should be sequentially adjusted for in separate models.

A: Revised as suggested (Please see page 4).

15. Q: In the conclusion the authors state ‘Our results also suggest that the impact of the adiposity trajectory on cardiometabolic profile is mediated by concurrent weight status.’ Is this a reference to tables 4 & 5 model 3? Could this association not also be mediated by age as age is included in model 3 along with weight status?

A: Revised as suggested, we have separated age and weight status in model 2 and model 3.

16. Q: Figures 1 & 2: I can’t see the figure legends anywhere in the submitted manuscript? These should be provided.

A: Revised as suggested

17. Q: Tables 2 & 3: the formatting is not quite right and makes the heading difficult to read.

A: Revised as suggested

18. Q: Discussion

The discussion is not well written and as such it is quite hard to follow the discussion points. The findings are discussed fairly superficially, more insight into potential physiological underpinnings for the different F2MR trajectories is necessary. For example, why might the high-stability trajectory group be associated with hypertension in men, independent of concurrent age and weight status? There is repetition of statements throughout the discussion which is unnecessary. 

A: Revised as suggested

19. Q: Consider opening the discussion with a summary paragraph of the key results and why they are important/novel.

A: Revised as suggested

20. Q: Paragraph 2: ‘dyslipidaemia was mainly affected by concurrent weight status’ I don’t think you can draw this conclusion without separating age/weight adjustment in model 3.

A: Revised as suggested (please see page 11)

21. Q: The limitation section should also include the high number of participants who fall into the low stability group and low numbers in the other groups and discussion of potential ethnic differences in body composition trajectories.

A: Revised as suggested (Please see Page)

---

## [Decision Letter · Decision Letter 1]

28 Mar 2021

PONE-D-20-38516R1

Long-term Changes in Body Composition and their Relationships with Cardiometabolic Risk Factors: A Population-Based Cohort Study

PLOS ONE

Dear Dr. Liu,

Thank you for submitting your manuscript to PLOS ONE. After careful consideration, we feel that it has merit but does not fully meet PLOS ONE’s publication criteria as it currently stands. Therefore, we invite you to submit a revised version of the manuscript that addresses the points raised during the review process.

We look forward to receiving your revised manuscript.

Kind regards,

Y Zhan

Academic Editor

PLOS ONE

Reviewers' comments:

Reviewer's Responses to Questions

**Comments to the Author**

1. If the authors have adequately addressed your comments raised in a previous round of review and you feel that this manuscript is now acceptable for publication, you may indicate that here to bypass the “Comments to the Author” section, enter your conflict of interest statement in the “Confidential to Editor” section, and submit your "Accept" recommendation.

Reviewer #2: (No Response)

2. Is the manuscript technically sound, and do the data support the conclusions?

Reviewer #2: No

3. Has the statistical analysis been performed appropriately and rigorously? 

Reviewer #2: I Don't Know

4. Have the authors made all data underlying the findings in their manuscript fully available?

Reviewer #2: Yes

5. Is the manuscript presented in an intelligible fashion and written in standard English?

Reviewer #2: Yes

6. Review Comments to the Author

Reviewer #2: The English language has largely been improved however I still do not think this paper is very well written. There are some outstanding and additional issues:

Introduction:

Line 45/46: “Millions of deaths can be ascribed to obesity worldwide[3], because it is associated with the development of cardiometabolic risk factors, such as dyslipidemia, diabetes, and hypertension[4, 5].”

This is not quite accurate. Dyslipidaemia, T2DM and hypertension are states of cardiometabolic disruption which increase the risk of adverse cardiovascular events.

Line 46/47: “Fat mass is used to evaluate obesity[6]”

In ref #6 they defined obesity using a BMI cut-off, so I don’t think this statement makes sense.

Line 61/62: “To date, few studies have characterized the trajectories of body composition and their associations with the development of cardiometabolic risk factors.”

I would say quite a large number of studies have tracked changes in body composition in longitudinal studies, however, I am unsure if this has previously been done in a Chinese population? Or if it has been done using the estimation of F2FFMR highlighting the importance of this work. Any more specific previous work on this should be referenced.

Methods:

(Q5&6) It is still not clear to me, from the written description and from Figure 1, how the authors arrived at the final sample size. What are the numbers in the box at the top right corner of fig 1?

“Those for whom data from at least two rounds of the survey were available were included.” How can you have a trajectory if you only have 2 measurements and such a wide age-range in this study sample? this should be explained, a supplementary information file might be helpful if the description is too lengthy for the main manuscript.

When were the cardiometabolic outcomes measured (at which visit)?

Was waist-hip ratio measured in this study?

(Q8) This is not true. Type 1 Diabetes develops in childhood but is still present into adulthood.

Results:

Table 5, statistically significant ORs not highlighted bold, as was done in Table 4.

Discussion:

Lines 247-250: “The effect of F2FFMR on cardiometabolic risk is likely to accumulate slowly, and high F2FFMR in early life might have an impact on cardiometabolic health in later life.”

It is not clear if these statements are based on evidence presented in previously published work? Or are speculation.

Line 256/256: “Study of the F2FFMR indicates that the adverse effects of high fat mass can be offset by the protective effects of high fat-free mass.”

Line 267/268: “The present findings also suggest that the impact of F2FFMR trajectory on cardiometabolic outcomes is mediated by changes in body mass.”

Is it ‘changes in body mass’ or just ‘current weight’? I think you should revise your interpretation of your results.

Line 270: do you mean the ‘high stability’ group or do you mean ‘highly stable’?

Line 294: I think the study sample from a Chinese only population living in China could be considered a strength of your study, given the previously published data from other parts of the world. There are other strengths that merit discussion as well.

Line299/300: “Our results also suggest that the association between F2FFMR trajectory and the cardiometabolic profile is mediated by body mass.”

If the associations are all mediated by current body mass status, then why do we need to know about the trajectories? It looks like your data do not fully support this statement: in women (table 5), dyslipidemia/diabetes/hypertension are all still higher in the ‘High Stability’ group despite adjustment for weight in 2009?

7. PLOS authors have the option to publish the peer review history of their article (what does this mean?). If published, this will include your full peer review and any attached files.

Reviewer #2: No

---

## [Author Response · Author response to Decision Letter 1]

4 Apr 2021

Response to the Reviewer’s Comments

Dear editor and reviewer,

We are grateful for the thoughtful reviews provided by the external referees and valuable comments from the editors. We have carefully addressed each recommendation, and the manuscript is considerably stronger as a result of the modifications outlined below.

The modified text is highlighted in tracked copy in the revised manuscript.

Question 1:

Introduction:

Line 45/46: “Millions of deaths can be ascribed to obesity worldwide [3], because it is associated with the development of cardiometabolic risk factors, such as dyslipidemia, diabetes, and hypertension[4, 5].”

This is not quite accurate. Dyslipidaemia, T2DM and hypertension are states of cardiometabolic disruption which increase the risk of adverse cardiovascular events.

Answer 1：Thank you for the question. we have revised as suggested. Please see line 45, page 3.

Question 2:

Line 46/47: “Fat mass is used to evaluate obesity[6]”

In ref #6 they defined obesity using a BMI cut-off, so I don’t think this statement makes sense.

Answer 2：Thank you for the question. we have revised as suggested. Please see line 47, page 3.

Question 3:

Line 61/62: “To date, few studies have characterized the trajectories of body composition and their associations with the development of cardiometabolic risk factors.”

I would say quite a large number of studies have tracked changes in body composition in longitudinal studies, however, I am unsure if this has previously been done in a Chinese population? Or if it has been done using the estimation of F2FFMR highlighting the importance of this work. Any more specific previous work on this should be referenced.

Answer 3：Thank you for the question. we have revised as suggested.

Please see line 62 to 66, page 3 to 4.

Question 4:

Methods:

(Q5&6) It is still not clear to me, from the written description and from Figure 1, how the authors arrived at the final sample size. What are the numbers in the box at the top right corner of fig 1?

Answer 4：Thank you for the question, we have revised as suggested. Please see figure 1.

We have revised figure 1 and make it clearer. In this study, we included 6 waves from 1993 to 2009 of the cohort. Participants we chose based on the wave 6 in 2009 which had blood biochemical results. There were 104, 63, 2, 62 new participants entered into the cohort in 1997, 2000, 2004 and 2006, respectively. The final sample size was 3013, which comprised of 2782 entered in 1993 and 231 new entered in 1997 to 2006.

Question 5:

“Those for whom data from at least two rounds of the survey were available were included.” How can you have a trajectory if you only have 2 measurements and such a wide age-range in this study sample? this should be explained, a supplementary information file might be helpful if the description is too lengthy for the main manuscript.

Answer 5：Thank you for the question. 

The formation of trajectory trend was based on all the data involved in the analysis. The procedure of model fitting was as: Firstly, when all the data was imported in the Mplus, linear models and quadratic models were fitted. It showed that quadratic models had much smaller Bayesian information criteria (BIC) values across all models. Therefore, further analyses were based on the quadratic curve assumption and 4 classes in men, 3 classes in women. After the models and classes were determined, everyone was classified into a trajectory class.

To maximize the use of the data, we include 2 measurements in this study as other researchers did in theirs studies. In this study, 2 measurements of F2FFMR was only 4.8% (145/3013). After we exclude the data of 2 measurements, there was no obvious change in Figure 2. 

1. Body mass index trajectories during the first year of life and their determining factors. American Journal of Human Biology. DOI: 10.1002/ajhb.23188

2. Body mass index trajectories during infancy and pediatric obesity at six years. Annals of Epidemiology.10.1016/j.annepidem.2017.10.008

Question 6:

When were the cardiometabolic outcomes measured (at which visit)?

Was waist-hip ratio measured in this study?

Answer 6：Thank you for the question, we have revised as suggested. The cardiometabolic outcomes were measured in wave 6 in 2009.There was no hip measure in this study. Please see page 6.

Question 7:

(Q8) This is not true. Type 1 Diabetes develops in childhood but is still present into adulthood.

Answer 7：Thank you for the information. I did some literatures review and found that China is one of the countries with the lowest incidence of type 1 diabetes. Although type 1 diabetes tends to develop in children, most of the new cases are diagnosed in adults. 

Weng J, Zhou Z, Guo L, Zhu D, Ji L, Luo X, Mu Y, Jia W; T1D China Study Group. Incidence of type 1 diabetes in China, 2010-13: population based study. BMJ. 2018 Jan 3;360:j5295. doi: 10.1136/bmj.j5295. 

In this study, the data was unavailable, we could not distinguish between type 1 diabetes and type 2 diabetes. There was no population-based data about the prevalence of type 1 diabetes in China, and it was 0.5% in US. So, the majority of diabetes in our study were T2DM. There might be a slight impact on our results.

Xu G, Liu B, Sun Y, Du Y, Snetselaar LG, Hu FB, Bao W. Prevalence of diagnosed type 1 and type 2 diabetes among US adults in 2016 and 2017: population based study.BMJ.2018,362:k1497.doi: 10.1136/bmj.k1497.

Question 8:

Results:

Table 5, statistically significant ORs not highlighted bold, as was done in Table 4.

Answer 8：Thank you for the question, we have revised as suggested. Please see page 14 and 15.

Question 9:

Discussion:

Lines 247-250: “The effect of F2FFMR on cardiometabolic risk is likely to accumulate slowly, and high F2FFMR in early life might have an impact on cardiometabolic health in later life.”

It is not clear if these statements are based on evidence presented in previously published work? Or are speculation.

Answer 9: Thank you for the question, we have revised as suggested. Please line 252 to 255, page 16. 

Question 10:

Line 256/256: “Study of the F2FFMR indicates that the adverse effects of high fat mass can be offset by the protective effects of high fat-free mass.”

Line 267/268: “The present findings also suggest that the impact of F2FFMR trajectory on cardiometabolic outcomes is mediated by changes in body mass.”

Is it ‘changes in body mass’ or just ‘current weight’? I think you should revise your interpretation of your results.

Answer 10: Thank you for the question, we have revised as suggested. It should be current body mass.

Question 11:

Line 270: do you mean the ‘high stability’ group or do you mean ‘highly stable’?

Answer 11: Thank you for the question, we have revised as suggested. Please see page 17.

Question 12: 

Line 294: I think the study sample from a Chinese only population living in China could be considered a strength of your study, given the previously published data from other parts of the world. There are other strengths that merit discussion as well.

Answer 12: Thank you for your advice. We added some points in it.

Question 13:

Line299/300: “Our results also suggest that the association between F2FFMR trajectory and the cardiometabolic profile is mediated by body mass.”

If the associations are all mediated by current body mass status, then why do we need to know about the trajectories? It looks like your data do not fully support this statement: in women (table 5), dyslipidemia/diabetes/hypertension are all still higher in the ‘High Stability’ group despite adjustment for weight in 2009?

Answer 13: Thank you for the question, we have revised as suggested. Please see page 18.

---

## [Decision Letter · Decision Letter 2]

28 Apr 2021

Long-term Changes in Body Composition and their Relationships with Cardiometabolic Risk Factors: A Population-Based Cohort Study

PONE-D-20-38516R2

Dear Dr. Liu,

We’re pleased to inform you that your manuscript has been judged scientifically suitable for publication and will be formally accepted for publication once it meets all outstanding technical requirements.

Kind regards,

Y Zhan

Academic Editor

PLOS ONE

Additional Editor Comments (optional):

Reviewers' comments:

Reviewer's Responses to Questions

**Comments to the Author**

1. If the authors have adequately addressed your comments raised in a previous round of review and you feel that this manuscript is now acceptable for publication, you may indicate that here to bypass the “Comments to the Author” section, enter your conflict of interest statement in the “Confidential to Editor” section, and submit your "Accept" recommendation.

Reviewer #2: (No Response)

2. Is the manuscript technically sound, and do the data support the conclusions?

Reviewer #2: Partly

3. Has the statistical analysis been performed appropriately and rigorously? 

Reviewer #2: I Don't Know

4. Have the authors made all data underlying the findings in their manuscript fully available?

Reviewer #2: Yes

5. Is the manuscript presented in an intelligible fashion and written in standard English?

Reviewer #2: No

6. Review Comments to the Author

Reviewer #2: The authors have addressed the comments sufficiently. The data is interesting and, to the best of my knowledge, the analysis is appropriate. English language used has been vastly improved however there are still small errors and places where the writing does not flow well. It is at the editors discretion whether these should be addressed.

7. PLOS authors have the option to publish the peer review history of their article (what does this mean?). If published, this will include your full peer review and any attached files.

Reviewer #2: No

---

## [Editor Report · Acceptance letter]

30 Apr 2021

PONE-D-20-38516R2 

Long-term Changes in Body Composition and their Relationships with Cardiometabolic Risk Factors: A Population-Based Cohort Study 

Dear Dr. Liu:

I'm pleased to inform you that your manuscript has been deemed suitable for publication in PLOS ONE. Congratulations! Your manuscript is now with our production department. 

Kind regards, 

on behalf of

Dr. Y Zhan 

Academic Editor

PLOS ONE